# An Automatic Sleep Stage Classification Algorithm Using Improved Model Based Essence Features

**DOI:** 10.3390/s20174677

**Published:** 2020-08-19

**Authors:** Huaming Shen, Feng Ran, Meihua Xu, Allon Guez, Ang Li, Aiying Guo

**Affiliations:** 1School of Mechatronics Engineering and Automation, Shanghai University, Shanghai 200444, China; ranfeng@shu.edu.cn (F.R.); mhxu@shu.edu.cn (M.X.); shulivia@shu.edu.cn (A.L.); gayshh@shu.edu.cn (A.G.); 2Faculty of Biomedical Engineering, Drexel University, Philadelphia, PA 19104, USA; guezal@drexel.edu

**Keywords:** EEG, sleep stage, wavelet packet, state space model

## Abstract

The automatic sleep stage classification technique can facilitate the diagnosis of sleep disorders and release the medical expert from labor-consumption work. In this paper, novel improved model based essence features (IMBEFs) were proposed combining locality energy (LE) and dual state space models (DSSMs) for automatic sleep stage detection on single-channel electroencephalograph (EEG) signals. Firstly, each EEG epoch is decomposed into low-level sub-bands (LSBs) and high-level sub-bands (HSBs) by wavelet packet decomposition (WPD), separately. Then, the DSSMs are estimated by the LSBs and the LE calculation is carried out on HSBs. Thirdly, the IMBEFs extracted from the DSSM and LE are fed into the appropriate classifier for sleep stage classification. The performance of the proposed method was evaluated on three public sleep databases. The experimental results show that under the Rechtschaffen’s and Kale’s (R&K) standard, the sleep stage classification accuracies of six classes on the Sleep EDF database and the Dreams Subjects database are 92.04% and 78.92%, respectively. Under the American Academy of Sleep Medicine (AASM) standard, the classification accuracies of five classes in the Dreams Subjects database and the ISRUC database reached 79.90% and 81.65%. The proposed method can be used for reliable sleep stage classification with high accuracy compared with state-of-the-art methods.

## 1. Introduction

Automatic sleep stage classification is an important research focus due to its importance for the study of sleep related disorders. There are currently two classification criteria for sleep stages. According to Rechtschaffen’s and Kale’s (R&K) recommendations, sleep stages can be divided into six stages: The Awake stage (Awa), rapid Eye Movement stage (REM), Sleep stage 1 (S1), Sleep stage 2 (S2), Sleep stage 3 (S3), Sleep stage 4 (S4) [1]. Another sleep stage classification standard was provided by the AASM. In this standard, there are five sleep stages: Awa, N1 (S1), N2 (S2), N3 (the merging of stages S3 and S4) and REM [2]. Usually, the detection of each sleep stage requires manual marking by professionals, which requires a lot of work and may produce erroneous markings. Therefore, it is imperative to study the method for automatic sleep stage classification.

According to the characteristics of the adopted features, currently commonly used automatic detection methods can be divided into the following two categories. The first is the method based on statistical features (such as spectral energy) extracted from the one-dimensional EEG signal. The other is the implicit features, which can be obtained by training deep-learning based classifiers. Hassan et al. computed various spectral features by Tunable-Q factor wavelet transform (TQWT) on sleep-EEG signal segments [3]. With the random forest classifier, they achieved accuracies of 90.38%, 91.50%, 92.11%, 94.80%, 97.50% for 6-stage to 2-stage classification of sleep states on the Sleep-EDF database. Diykh et al. adopted different structural and spectral attributes extracted from weighted undirected networks to automatically classify the sleep stages [4]. Kang et al. present a statistical framework to estimate whole-night sleep states in patients with obstructive sleep apnea (OSA)—the most common sleep disorder [5]. In this framework, they extracted 11 spectral features from 60903 epochs to estimate per-night sleep stages with a 5-state hidden Markov model. Abdulla et al. used graph modularity of EEG segments as the features to feed an ensemble classifier which achieved the accuracy of 93.1% with 20265 epochs from Sleep EDF database [6].

In [7], Ghimatgar et al. constructed a features pool by the relevance and redundancy analysis on the sleep EEG epochs. With a random forest classifier and a Hidden Markov Model, this method was evaluated on three public sleep EEG database scored according to R&K and AASM guidelines. They achieved overall accuracies in the range of (79.4–87.4%) and (77.6–80.4%) for six-stage (R&K) and five-stage (AASM) classification, respectively. Taran et al. proposed an optimized flexible analytic wavelet transform (OFAWT) to decompose EEG signals into band-limited basis or sub-bands (SBs) [8]. The experimental results yields classification accuracies for the classification of six to two sleep stages 96.03%, 96.39%, 96.48%, 97.56% and 99.36%, respectively. Sharma et al. computed the discriminatory features namely fuzzy entropy and log energy by the wavelet decomposition coefficients [9]. This approach yielded an accuracy of 91.5% and 88.5% for six-class classification task using small and large datasets, respectively. Hassan et al. extracted various statistical moment based features decomposed by the Empirical Mode Decomposition (EMD) and achieved a good performance on a small database [10]. They also decomposed EEG signal segments using Ensemble Empirical Mode Decomposition (EEMD) to extract various statistical moment based features and achieved 88.07%, 83.49%, 92.66%, 94.23% and 98.15% for 6-state to 2-state classification of sleep stages on Sleep-EDF database [11]. Sharma et al. adopted the Poincare plot descriptors and statistical measures which are calculated by the discrete energy separation algorithm (DESA) as the features [12]. Moreover, the classification accuracy of the two to six categories on 15136 epochs from the Sleep-EDF database was 98.02%, 94.66%, 92.29%, 91.13% and 90.02%, respectively.

Besides the conventional features extraction method, some researchers choose the convolutional neural network (CNN) to classify sleep stages with the time–frequency images which are converted by one-dimensional EEG signals. Zhang et al. converted EEG data to a time–frequency representation via Hilbert–Huang transform and employed an orthogonal convolutional neural network (OCNN) as the classifier [13]. They achieved a total classification accuracy of 88.4% and 87.6% on two public datasets, respectively. Similarly, Xu et al. employed multiple CNN on multi-channel EEG signals to classify the sleep stages [14]. Mousavi [15] directly fed the raw EEG signals to a deep CNN with nine layers followed by two fully connected layers, without involving feature extraction and selection. This method achieved the accuracy of 98.10%, 96.86%, 93.11%, 92.95%, 93.55% for two to six class classification. Long short-term memory (LSTM) is an artificial recurrent neural network (RNN) architecture used in the field of deep learning. It can not only process single data points (such as images), but also entire sequences of data (such as speech or EEG signal). Korkalainen et al. used a combined convolutional and LSTM neural network on the public database and achieved sleep staging accuracy of 83.7% with a single frontal EEG channel [16]. Michielli et al. proposed a novel cascaded RNN architecture based on LSTM for automated scoring of sleep stages on single-channel EEG signals [17]. The network performed four and two classes classification with a classification rate of 90.8% and 83.6%, respectively.

Most of the existing studies only adopted a few epochs or a single database when evaluating the performance of these method and some do not use the k-fold cross-validation, which will cause large fluctuations in the experimental results. Therefore, although the published researches have achieved positive results in automatic sleep stage classification, there is still a need for further validation and improvements to the existing methods. In this paper we proposed a novel IMBEFs extracted from LE and DSSM for automatically detecting the sleep stages with a high degree of accuracy. LE and DSSM are estimated from the two sets coefficients of LSBs and HSBs. The two sets coefficients are coming from the WPD of the sleep EEG epoch based on two wavelet bases separately. After comparing with various kinds of classifiers, the Bagged Trees was finally selected as the suitable classifier for this method to identify the sleep stages. In addition, experiments are conducted on three public sleep databases and the results are compared with state of the art published work in order to fully evaluate and validate the performance of the proposed method.

The paper is organized as follows: In Section 2, the experimental material and methodology of the proposed method are descripted in detail. Section 3 resents the experimental results. In Section 4, the results and findings of this paper are discussed. The conclusions of the paper are drawn in Section 5.

## 2. Materials and Methods

### 2.1. Sleep State Classes

According to the AASM and R&K standards, the classes of sleep stages can be divided into two to six classes. Moreover, under the AASM standard, it can be divided into two to five classes. The difference is that the N3 stage of AASM includes the S3 and S4 stages of the R&K standard. The detailed description of classes considered in this work are shown in Table 1 and Table 2.

### 2.2. Datasets

#### 2.2.1. Sleep EDF (S-EDF) Database

The S-EDF database have 197 whole-night Polysomnography (PSG) sleep recordings, containing EEG, EOG, chin EMG and event markers [18,19]. All the Hypnograms (sleep patterns) were manually scored by well-trained technicians according to the R&K criteria. In this study, 34 EEG recordings from 26 subjects aged 25 to 96 years are randomly selected.

#### 2.2.2. DREAMS Subjects (DRMS) Database

The DRMS Database consists of 20 whole-night PSG recordings coming from healthy subjects, annotated in sleep stages according to both the R&K criteria and the new standard of the AASM [20]. Data collected were acquired in a sleep laboratory of a Belgium hospital using a digital 32-channel polygraph (BrainnetTM System of MEDATEC, Brussels, Belgium). The sampling frequency was 200 Hz.

#### 2.2.3. ISRUC(Subgroup 3, ISRUC3) Database

The ISRUC3 database is the third subgroup of ISRUC database [21]. The data were obtained from human adults, including healthy subjects, subjects with sleep disorders and subjects under the effect of sleep medication. Each recording was randomly selected between PSG recordings that were acquired by the Sleep Medicine Centre of the Hospital of Coimbra University (CHUC).

The S-EDF database was only labeled under the R&K criteria. Moreover, the ISRUC3 database was only labeled by the AASM criteria. The DRMS database was not only labeled by R&K criteria but also the AASM criteria. The annotations of S-EDF database and DRMS database were produced visually by a single expert. The ISRUC3 database was scored by two experts and the label made by the second expert was used in this paper. The Pz-Oz channel of the S-EDF database is used according to the recommendations of various studies [3,4,5,6,7]. At the same time, for the DRMS database, as the researches [9,10,11,12] recommended, the Cz-A1 channel was adopted in this work. Moreover, for the ISRUC database, the C3-A2 channel is the best choice [7]. Table 3 lists the detailed information of the above three databases.

### 2.3. Method

Figure 1 shows the schematic outline of the proposed IMBEFs based sleep statge classification algorithm comprising preprocessing, wavelet package decomposition, locality energy calculation, state space models estimation, features extraction, classifier training and performance evaluation.

#### 2.3.1. EEG Data Preprocessing

Firstly, all the single-channel data will be extracted by the Matlab and EEGLAB [22] tools from the three database described previously. According to the prior work [5,6,7,8,9,10,11], the 0–35 Hz low pass filter can be used to eject the most of artifact. Once the dataset is filtered, it will be exported as one-dimensional vector without time information and saved as txt file which also can be denoted as the Formula (Equation 1).
(1)$$X=\left[x_{1},x_{2},\ldots ,x_{k},\ldots ,x_{M}\right],k\in \left[1,M\right],x_{k}\in R$$
where X is the vector containing the sampled EEG $x_{k}$ and where *M* is the length of vector.

Furthermore, we use a window of length *j* to divide the full data X across time without overlap. That is X is converted into ${\left[X_{1},X_{2},\ldots ,X_{i},\ldots ,X_{L}\right]}^{T}$ which can be described as (Equation 2).
(2)$$\begin{array}{cc} \left[\begin{array}{c} X_{1}\\ X_{2}\\ \vdots \\ X_{i}\\ \vdots \\ X_{L} \end{array}\right]= & \left[\begin{array}{cccc} x_{1} & x_{2} & \cdots & x_{j}\\ x_{j+1} & x_{j+2} & \cdots & x_{2j}\\ \vdots & \vdots & \cdots & \vdots \\ x_{(i-1)j+1} & x_{(i-1)j+2} & \cdots & x_{i\times j}\\ \vdots & \vdots & \vdots & \vdots \\ x_{(L-1)j+1} & x_{(L-1)j+2} & \cdots & x_{L\times j} \end{array}\right]\\ \; & L=\left\lfloor \frac{M}{j}\right\rfloor ,i\in \left[1,L\right] \end{array}$$
where $j=T_{e}\times F_{s}$. The $T_{e}$ is the length of each epoch. Moreover, the $F_{s}$ is the sampling frequency of the database. For the S-EDF database, the $T_{e}=30$ and the $F_{s}=100$, so the *j* is 3000. Moreover, for the ISRUC3 database, the $T_{e}=20$ and the $F_{s}=200$, so the *j* is 4000.

#### 2.3.2. Wavelet Package Decomposition

WPD is a powerful tool to analyze non-stationary EEG signals. In essence, WPD is a wavelet transform where the discrete-time signal is passed through more filters than the discrete wavelet transform, which can provide a multi-level time-frequency decomposition of signals [23]. Compared with discrete wavelet transform, WPD can provide more frequency resolutions. In the discrete wavelet transform, a signal is split into an approximation coefficient and a detail coefficient [24]. The approximation coefficient is then itself split into a second-level approximation coefficients and detail coefficients and the process is repeated. A wavelet packet function $\omega_{l,d}^{m}\left(q\right)$ is defined as (Equation 3):(3)$$\omega_{l,d}^{m}\left(q\right)=2^{l/2}\omega^{m}\left(2^{l}q-d\right)$$
where *l* and *d* are the scaling (frequency localization) parameter and the translation (time localization) parameter, respectively; m=0,1,2,… is the oscillation parameter.

Wavelet packet (WP) coefficients of the EEG epoch $X_{i}$ are embedded in the inner product of the signal with every WP function, denoted by $p_{l}^{i,m}\left(d\right),d=\ldots ,-1,0,1,\ldots$ and given below:(4)$$p_{l}^{i,m}\left(d\right)=\sum x^{i}\left(q\right)\omega_{l,d}^{m}\left(q\right)$$
where $p_{l}^{i,m}\left(d\right)$ denotes the *m*-th set of WPD coefficients at *l*-th scale parameter and *d* is the translation parameter. All frequency components and their occurring times are reflected in $p_{l}^{i,m}\left(d\right)$ through change in m,l,d. Each coefficient $p_{l}^{m}\left(d\right)$ measures a specific sub-band frequency content, controlled by scaling parameter *l* and oscillation parameter *m*. The essential function of WPD is the filtering operation through low-pass filter h(d) and high-pass filter g(d). For the *l*-th level of decomposition, there are $2^{l}$ sets of sub-band coefficients $C_{l,m}^{i}$, of length $j/2^{l}$. The wavelet packet coefficients of epoch $X_{i}$ are given as
(5)$$C_{l,m}^{i}=\left\{p_{l}^{i,m}\left(d\right)|d=1,2,...,j/2^{l}\right\}$$

It can be seen from the (Equation 5) that each node of the WP tree is indexed with a pair of integers (l,m), where *l* is the corresponding level of decomposition and *m* is the order of the node position in the specific level. Here, the level $l_{LE}$ and wavelet basis $\omega_{LE}$ of WPD on the epoch $X_{i}$ for LE calculation will be confirmed in the Section 3. Moreover, the wavelet basis $\omega_{DSSM}$ for DSSM will be confirmed in the same section.

#### 2.3.3. Locality Energy Calculation

The wavelet package energy $E_{l_{LE},m}^{i}$ at the *m*-th node on the level $l_{LE}$ of epoch $X_{i}$ can be defined as follows [25].
(6)$$E_{l_{LE},m}^{i}=\sum |p_{l}^{i,m}{\left(d\right)|}^{2}={\left|C_{l_{LE},m}^{i}\right|}^{2},m=\left\{1,2,\ldots ,2^{l_{LE}}\right\}$$

Then, the locality energy features (LEFs) of each Epoch can be defined as $\left\{E_{l_{LE},m}^{i}|m=1,2,\ldots ,2^{l_{LE}}\right\}$.

#### 2.3.4. Dual State Space Models Estimation

As we have described before, after the wavelet packet decomposition, the low-level (the first level) coefficients will be used to estimate the dual state space models which can denoted by the difference Equation (Equation 7).
(7)$$\left\{\begin{array}{c} u_{k+1}=Au_{k}+Ke_{k}\\ y_{k}={\mathrm{Bu}}_{k}+e_{k} \end{array}\right.$$

The $y_{k}\in C_{1,m}^{i}$ is the coefficient at instant k∈[1,2,…,j/2]. Vector $u_{k}\in R^{n\times 1}$ is the state vector of process at discrete time instant *k* and contains the numerical value of *n* states. Matrix $A\in R^{n\times n}$ is the dynamical system matrix. $K\in R^{n\times 1}$ is the steady state Kalman gain. $B\in R^{1\times n}$ is the output matrix, which describes how the internal state is transferred to the outside world in the observations $y_{k}$. The $e_{k}\in R$ denotes zero mean white noise.

With the traditional subspace algorithm such as N4SID, the matrix $\hat{A}$, $\hat{B}$, $\hat{K}$ of the state space model of dynamic system can be estimated [26]. In this paper, the order $n_{DSSM}$ of dual state space models will be determined by the experiments in the Section 3. Moreover, the parameter matrixes of state space model estimated by the first level wavelet coefficients $C_{1,m}^{i}$ can be expressed as
(8)$$\begin{array}{cc} \; & {\hat{A}}_{1,m}^{i}=\left[\begin{array}{ccc} a_{1,1}^{i,m} & \cdots & a_{1,n_{DSSM}}^{i,m}\\ \vdots & \cdots & \vdots \\ a_{n_{DSSM},1}^{i,m} & \cdots & a_{n_{DSSM},n_{DSSM}}^{i,m} \end{array}\right]\\ \; & {\hat{B}}_{1,m}^{i}=\left[b_{1}^{i,m}\;b_{2}^{i,m}\cdots \;b_{n_{DSSM}}^{i,m}\right]\\ \; & {\hat{K}}_{1,m}^{i}={\left[k_{1}^{i,m}\;b_{2}^{i,m}\cdots \;k_{n_{DSSM}}^{i,m}\right]}^{T}\\ \; & i\in \left[1,L\right],m=\left\{1,2\right\} \end{array}$$

Then the DSSM $S_{i}$ of the $X_{i}$ can be defined as:(9)$$S_{i}=\left[\begin{array}{c} s_{1}^{i}\\ s_{2}^{i} \end{array}\right]=\left[\begin{array}{ccccccccc} a_{1,1}^{i,1} & \cdots & a_{n_{DSSM},n_{DSSM}}^{i,1} & b_{1}^{i,1} & \cdots & b_{n_{DSSM}}^{i,1} & k_{1}^{i,1} & \cdots & k_{n_{DSSM}}^{i,1}\\ a_{1,1}^{i,2} & \cdots & a_{n_{DSSM},n_{DSSM}}^{i,2} & b_{1}^{i,2} & \cdots & b_{n_{DSSM}}^{i,2} & k_{1}^{i,2} & \cdots & k_{n_{DSSM}}^{i,2} \end{array}\right]$$

So, the parameters extracted from DSSM here is called DSSM Features (DSSMFs) can be defined as $DSSMFs=\left[s_{1}^{i}\;s_{2}^{i}\right]$.

#### 2.3.5. IMBEFs Construction

According to the previously calculated LEFs $E_{l_{LE},m}^{i}$ and the parameters $S_{i}$ of the DSSM, the features IMBEFs of epoch $X_{i}$ here are given by
(10)$$F_{DSSM}^{i}=\left[E_{l_{LE},1}^{i}\;\cdots \;E_{l_{LE},2^{l_{LE}}}^{i}\;s_{1}^{i}\;s_{2}^{i}\right]$$

The feature dimension can be calculated by the Equation (Equation 11).
(11)$$Dim_{DSSM}=2^{l_{LE}}+2\left(n_{DSSM}^{2}+2\times n_{DSSM}\right)$$

Here, the general form of features extracted from LE and multiple state space models (MSSM) which are estimated by the $l_{MSSM}$-th level WPD coefficients can be depicted as Equation (Equation 12).
(12)$$F_{MSSM}^{i}=\left[E_{l_{LE},1}^{i}\;\cdots \;E_{l_{LE},2^{l_{LE}}}^{i}\;s_{1}^{i}\;\cdots \;s_{2^{l_{MSSM}}}^{i}\right]$$

The dimension of the $F_{MSSM}^{i}$ can be calculated by
(13)$$Dim_{MSSM}=2^{l_{LE}}+2^{l_{MSSM}}\left(n_{MSSM}^{2}+2\times n_{MSSM}\right)$$
where $n_{MSSM}$ is the order of MSSM. Usually, the $n_{MSSM}$ range from 5 to 10. Assume the $n_{MSSM}=5$, $Dim_{DSSM}=2^{l_{LE}}+2^{l_{MSSM}}\times 40$. Then if $l_{MSSM}>2$, the $Dim_{MSSM}$ will be too large. So the $l_{MSSM}$ is set to 1 in this paper, which means there are two state space models.

## 3. Experiments and Results

In this section, there are four experimental parts. The first is the experiment for selecting a suitable classifier among several candidate classifiers. Then is the determination of the most suitable wavelet basis $\omega_{DSSM}$ and model order $n_{DSSM}$ for DSSM estimation. Next is the selection of the wavelet basis $\omega_{LE}$ and the level $l_{LE}$ for the LE calculation. Finally, the test experiment will be conducted on the S-EDF database and ISRUCS3 database with the $\omega_{DSSM}$, $\omega_{LE}$, $n_{DSSM}$ and $l_{LE}$ determined according to the previous experiments.

In the process of selecting these parameters, the DRMS database was adopted for testing under the both R&K and AASM standards. There are several conventional verification strategies, including k-fold cross-validation, leave one-subject-out cross-validation (LOOCV) and corss-dataset validation, etc. In this paper, many commonly-used databases are adopted to verify the performance of the algorithm, in which the S-EDF database and the DRMS database contains lots of subjects. However, some subjects contained in these database possess unevenly distributed samples, which means the incomplete sleep stages. Consequently, the 10-fold cross-validation method would be more suitable for the performance verification in this research. In 10-fold cross-validation, the original sample is randomly partitioned into 10 equal size subsamples. Of the 10 subsamples, a single subsample is retained as the validation data for testing the model and the remaining nine subsamples are used as training data. The cross-validation process is then repeated 10 times, with each of the 10 subsamples used exactly once as the validation data. The 10 results from the folds can then be averaged to produce a single estimation. The advantage of this method is that all observations are used for both training and validation and each observation is used for validation exactly once. The accuracy (ACC) and Cohen’s Kappa Coefficient (Kappa) are computed to evaluate the overall classification performance.
(14)$$ACC=\frac{TP+TN}{TP+TN+TN+FN}\times 100\%$$
(15)$$Kappa=\frac{ACC-p_{e}}{1-p_{e}}$$
where TP, TN, FP and FN represent the number of true positive, true negative, false positive and false negative examples respectively. And $p_{e}$ is the hypothetical probability of agreement by chance.

### 3.1. Classifier Comparison and Selection

In this section, an algorithm is designed to search the best classifier for the method proposed in this paper. The detailed steps are shown in the Algorithm 1 below. In this algorithm, according to the distribution and characteristic of the samples, the candidate classifiers are including Linear Discriminant, Quadratic Discriminant, Quadratic SVM, Fine KNN, Bagged Trees and RUSBoosted Trees. The candidate wavelet bases include the db1, db2, db3, db4, db5, db6, db8, db16, db32, sym2, sym8, sym16, coif1, coif3 and dmey. The order of DSSM range from 5 to 10. Here only the DSSMFs are used for training and validation.

Table 4 shows the experiment results of Algorithm 1. As can be seen from Table 4, the Bagged Trees is the optimal classifier in the classification of two to six classes. At the meantime, the corresponding order of DSSM is 6. In addition, in the two class classification, the optimal wavelet basis is sym2; the others, however, are db1. Furthermore, the comparison of different classifiers in two classes classification under the condition of $n_{DSSM}=6$ are listed in Table 5. It can be seen from Table 5 that the accuracy of sym2 is 95.79%, which is a little higher than the 95.71% of db1 and 95.72% of db2. Therefore, considering the results in Table 4 and Table 5, the Bagged Trees will be used as the classifier for subsequent experiments.
**Algorithm 1:** Search the Optimal Classifier.
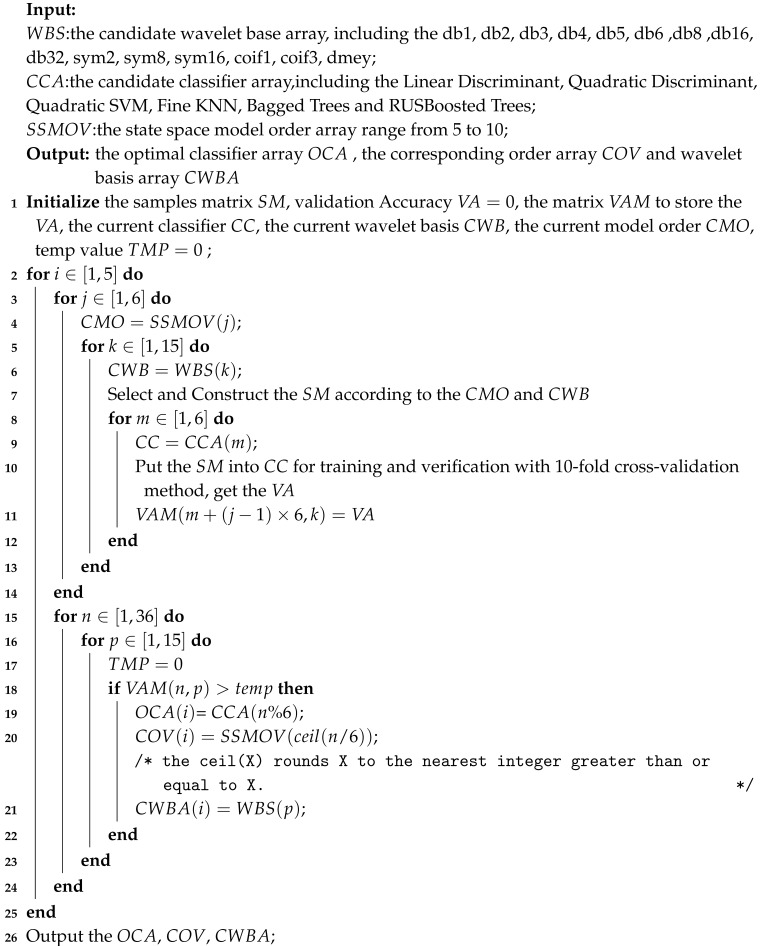


### 3.2. Wavelet Basis Comparison and Selection

After the classifier is determined, the model order $n_{DSSM}$ and wavelet basis $\omega_{DSSM}$ should be further confirmed through the grid search method. This process can be seen in the step 1 of the Figure 2. The candidate wavelets include db1, db2, db3, db4, db5, db6, db8, db16, db32, sym2, sym8, sym16, coif1, coif3 and dmey. The candidate model order is 5 to 10. The Following Table 6, Table 7, Table 8, Table 9, Table 10, Table 11, Table 12, Table 13 and Table 14 are the experiments results of the DRMS database without LEFs, in which the highest accuracy values are highlighted in bold.

From Table 6, Table 7, Table 8, Table 9 and Table 10, we can see that under the R&K standard, when the order of the DSSM is 6 and the wavelet basis is selected as db1, the classification accuracy for three to six classes can reach the highest. When the wavelet basis is selected as sym2, the accuracy of the two classes is the highest. Through further analysis, it can be seen that in the results of two class classification, the difference between the accuracy of the db1 and the highest is very small.

As can be seen from Table 11, Table 12, Table 13 and Table 14, when the order $n_{DSSM}$ is 6, the highest classification accuracy can be obtained in two to five classes sleep state classification. Moreover, in the three to five classes classifications, when the wavelet basis is db1, the highest classification accuracy can be achieved. In the two classes of sleep classification, when the wavelet base is db1, the accuracy is 0.14% lower than the highest accuracy. Combining the classification results of the above tables, in order to facilitate subsequent calculations, the db1 was uniformly used as the wavelet basis for DSSM estimation and the model order of DSSM adopts 6.

Then, the wavelet basis $\omega_{LE}$ and level $l_{LE}$ which are required to calculate LE should be further determined according to the experimental results in the next step. That is, on the basis of the features previously extracted from the DSSM, LEFs will be added which have been shown in the Step 2 of the Figure 2. Table 15, Table 16, Table 17, Table 18 and Table 19 are the classification accuracies of 2–6 classes under the R&K standard, in which the highest accuracy values are highlighted in bold.

As can be seen from Table 15, Table 16, Table 17, Table 18 and Table 19, when $l_{LE}=5$, the $\omega_{LE}$ is db4, the accuracy of two, four and six classes is the highest. Moreover, when the $\omega_{LE}$ is set to the db5 and db3, the classification accuracy of three and five classes can reach the highest respectively. The Table 20 is the confusion matrix of six classes sleep state classification on DRMS database with IMBEFs under the R&K standard. As shown in the Table 20, the sensitivity of Awa, REM, S1, S2, S3 and S4 are 93.68%, 81.16%, 14.37%, 89.29%, 25.71% and 77.99%, respectively. Moreover, the overall accuracy of the six classes classification is 78.92%.

Table 21, Table 22, Table 23 and Table 24 show the classification accuracy of 2–5 classes with LEFs on the DRMS database under the AASM, in which the highest accuracy values are highlighted in boldface. As can be seen from these tables, after adding LEFs, the accuracy of each classification has been greatly improved. Among them, the highest accuracy can be obtained when using the LEFs extracted from the 5 level WPD and there are three corresponding wavelet bases, which are db1, db2 and db4. When the wavelet basis is selected as db4, the accuracy of two classes and four classes can reach the highest. In addition, the accuracy of three and five classes are 88.22% and 79.90% respectively, which is not much different from 88.26% and 79.97% of the corresponding highest classification accuracy. Therefore, the parameter of $l_{LE}$ will be set as 5 and $\omega_{LE}$ will be set as db4 in this paper.

The confusion matrix of five classes sleep state classification is listed in the Table 25. As can be seen in this table, the overall accuracy is 79.90%. The sensitivity of Awa, REM, N1, N2, N3 are 92.89%, 81.22%, 17.57%, 85.52% and 78.79%. Furthermore, the receiver operating characteristic (ROC) curve of the classifier trained by this dataset with the confirmed parameter is shown in Figure 3.

As can be seen in the Figure 3, when the positive samples is Awa, the true positive rate is 0.93 and the false positive rate is 0.05. In addition, when the positive samples are REM, N2 and N3, the corresponding positive sample rates are 0.81, 0.86 and 0.79. When the positive samples are N1, the area under the curve (AUC) area is only 0.18. Moreover, the issue of low classification accuracy of S1(N1) will be discussed in the Section 4.

### 3.3. Experiments on S-EDF and ISRUC3 Database

After experiments on the DRMS database, through the comprehensive comparison and selection, the classifier is selected as the Bagged Tress, $n_{DSSM}$ is set to 6, $l_{LE}$ is set to 5, $\omega_{DSSM}$ is set to db1 and $\omega_{LE}$ is set to db4. In order to further evaluate the performance of the method proposed in this paper, we will use these parameters to conduct experiments on the S-EDF database and the ISRUC3 database.

The classification accuracy and Cohen’s Kappa Coefficients of the 2–6 classes on the S-EDF database are shown in Table 26. Furthermore, the confusion matrix of six class classification is listed for further analysis in Table 27.

Similarly, the method proposed in this paper was also tested on the ISRUC3 database. The experimental results are shown in the following Table 28 and Table 29.

As can be seen from Table 28, the classification accuracies of two to five classes are 96.18%, 90.54%, 84.68% and 81.65%, respectively. In the five class classification, the sensitivity of Awa, REM, N1, N2, N3 are 90.31%, 83.36%, 57.70%, 81.12% and 87.50%, respectively.

## 4. Discussion

Table 30 shows the comparison of the classification accuracy from two to six classes of the various published method and the method proposed in this paper on the DRMS database under the R&K standard.

As can be seen from the Table 30 above, when the only DSSMFs is used, the method proposed in this paper has a certain improvement in accuracy compared with the others. After adding LEFs on the basis of DSSMFs, the classification accuracies of two to six classes are improved by 1.27%, 1.02%, 1.27%, 1.38% and 0.72% compared with our previous study [27].

It can be seen from Table 31 that the method proposed in this paper has a certain improvement in the sleep stage classification of 3–5 classes on the DRMS database compared with the current existing methods. The N1 sensitivity of this method on the DRMS database is 17.57%, which is higher than 14.3% of Ghimatgar [7]. Moreover, Table 32 is the accuracy comparison of various published methods on S-EDF database.

It can be seen from Table 32 that when a large number of samples are used, the accuracy is also improved compared with other published methods. Among them, the accuracy for the classification of four classes is 93.87%, while the Sharma [28] is 92.1% and the Shen [27] is 93.0%. In the classification of two classes, Abdulla et al. [6] has the highest accuracy of 93%; however, the number of epoch they used is only 23806. The sensitivity of S1 in this paper is 19.32%, which is higher than 18.3% of Ghimatgar [7] and 15.9% of Shen [27].

The experiments results of the proposed method on ISRUC3 database are also compared with other methods, which can be seen in the following Table 33.

As can be seen from the Table 33, compared with Ghimatgar [7], the detection accuracy of two and three classes is improved by more than 2 points. The sensitivity of S1 in Table 29 is 57.70%, which is higher than 33% of Ghimatgar [7]. Furthermore, the Cohen’s kappa Coefficient is also much higher than Ghimatgar [7].

It should be noted that the classification of S1 which is an enormous challenge to all of the published method. From neurophysiological standpoint, S1(N1) is a transition phase and is a mixture of wakefulness and sleep resulting in similarity with the neural oscillations of S1 and Awa. In REM state, the cortex shows 40–60 Hz gamma waves as it does in waking. So the S1 state is often misclassified as REM or Awa state during the visual inspection by experts [3,11]. This is why many of the S1 epochs are misclassified as REM, Awa or S2 stages in this work. In addition, with different databases, the classification accuracy of S1 (N1) are also different. The detection accuracy of N1 on the ISRUC3 database reached 57.7%; on the DRMS database and the S-EDF database, however, it is less than 20%. This is also related to the different proportions of S1 stages in each database. Under the same AASM standard, on the ISRUC3 database, the S1 accounted for 12.65%; however, on the DRMS database, the S1 accounted for only 7.3%. Furthermore, under the R&K standard, the sensitivity of S3 on the S-EDF and DRMS databases is low, only 46.11% and 25.71%, respectively. The reason relate to this phenomenon rely mainly on that the S3 is a transition phase of S2 and S4. Thus the further research should be conducted to improve the S3 detection accuracy. Moreover, as can be seen in Table 20, a large number of S3 is misclassified as S2 and the other large part is misclassified as S4. Similarly, in Table 27, almost half of S3 epochs are misclassified as S2 and a small part are misclassified as S4. In addition, when under the AASM standard, after combining the S3 and S4 into N3, the sensitivity of N3 has been improved. As shown in Table 25, only 761 epochs of N3 were misclassified as N2; however, in Table 20, 1022 epochs of S3 were misclassified as S2 and 231 epochs of S4 were misclassified as S2. Therefore, the AASM standard is more suitable for guiding the researchers to annotate the sleep stages than the R&K standard.

## 5. Conclusions

In this paper, a novel IMBEF based automatic sleep stage classification method is proposed. Moreover, a grid search strategy was presented to determine a suitable model order $n_{DSSM}$ and a wavelet basis $\omega_{DSSM}$ for estimating the DSSM among 15 candidate wavelets and 6 candidate model orders. With the same search strategy, a proper wavelet basis $\omega_{LE}$ and the WPD level $l_{LE}$ for LE calculation are determined under 15 candidate wavelets and multilevel decomposition. The fused IMBEFs extracted from the DSSM and LE would be used as the input features of the suitable classifier which can be selected by comparing a variety of classifiers’ experiment results. In order to precisely verify the performance of the proposed IMBEF based automatic sleep stage classification method, experiments were carried out on three public databases. The comparison results with other state-of-the-art methods show that the proposed algorithm can achieve higher accuracy.

We demonstrated in this paper measurable improvements in automatic sleep stage classification, providing better understanding and diagnostic of the sleep phenomenon, clearly essential in medical, wellness and other fields.

## Figures and Tables

**Figure 1 sensors-20-04677-f001:**
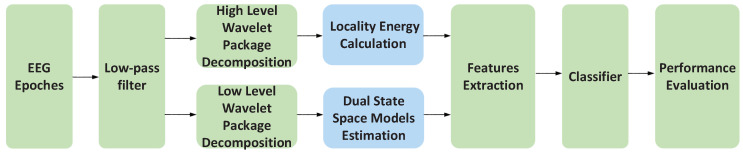
A schematic outline of the proposed improved model based essence features (IMBEFs) based sleep stage classification algorithm.

**Figure 2 sensors-20-04677-f002:**
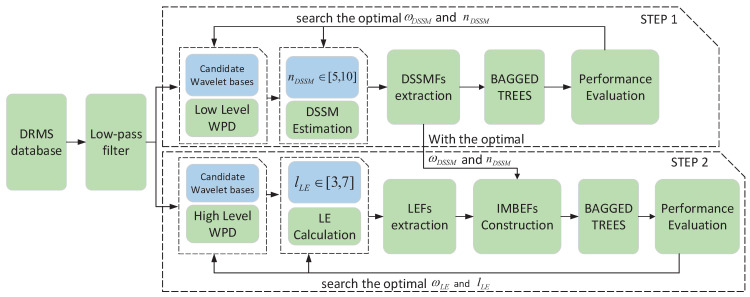
The diagram of the parameter optimization process.

**Figure 3 sensors-20-04677-f003:**
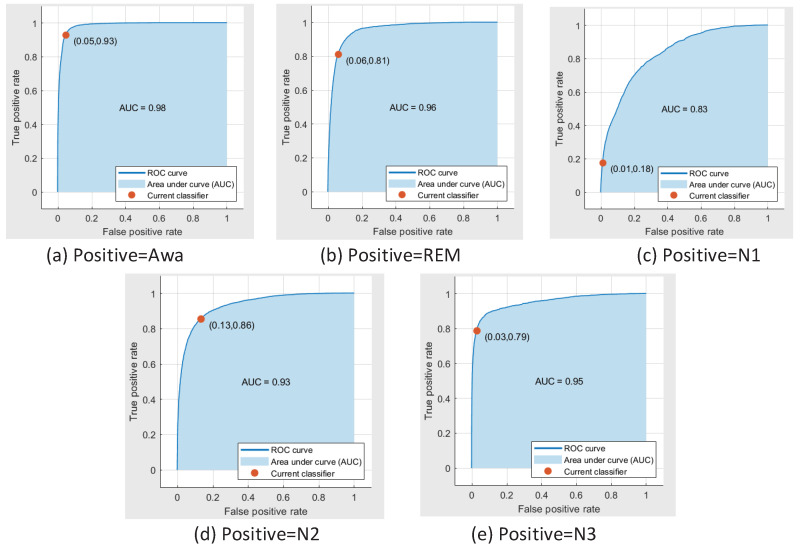
The ROC curve of the classifier to classify the five classes of DRMS database under the AASM standard.

**Table 1 sensors-20-04677-t001:** The class description considered in this work under the Rechtschaffen’s and Kale’s (R&K) standard.

Classes	6 Classes	5 Classes	4 Classes	3 Classes	2 Classes
Stages	Awa vs. REM vs. S1 vs. S2 vs. S3 vs. S4	Awa vs. REM vs. S1 vs. S2 vs. S3, S4	Awa vs. REM vs. S1, S2 vs. S3, S4	Awa vs. REM vs. NREM (S1, S2, S3, S4)	Awa vs. Asleep (REM, S1, S2, S3, S4)

**Table 2 sensors-20-04677-t002:** The class description considered in this work under the American Academy of Sleep Medicine (AASM) standard.

Classes	5 Classes	4 Classes	3 Classes	2 Classes
Stages	Awa vs. REM vs. N1 vs. N2 vs. S3, S4	Awa vs. REM vs. N1, N2 vs. N3	Awa vs. REM vs. NREM (N1, N2, N3)	Awa vs. Asleep (REM, N1, N2, N3)

**Table 3 sensors-20-04677-t003:** The specification of the electroencephalograph (EEG) databases included in this study.

Scoring Manual	R&K Criteria		AASM Criteria	
dataset name	S-EDF database	DRMS database	DRMS database	ISRUC3 database
Epoch length(Seconds)	30	20	30	20
Number of subjects	26	20	20	10
Recoding Files	34	20	20	10
Age	25–96	20–65	20–65	30–58
Gender(male-female)	17–17	4–16	4–16	9–1
Sampling frequency (Hz)	100	200	200	200
EEG channel	Pz–Oz	Cz–A1	Cz–A1	C3–A2
Stage	Number of epochs			
Awa	7,3835	5601	3559	1702
REM	6744	4555	3019	1238
S1(N1)	3017	1788	1480	1123
S2(N2)	1,7249	1,3274	8251	2850
S3(N3)	2288	2112	3956	1976
S4	1510	3071	–	–
Total Number of Epochs	10,4643	3,0401	2,0265	8889

**Table 4 sensors-20-04677-t004:** The outputs of Algorithm 1.

Classes	Optimal Classifier	$\omega_{\mathrm{DSSM}}$	$n_{\mathrm{DSSM}}$	Accuracy(%)
2	Bagged Trees	sym2	6	95.79
3	Bagged Trees	db1	6	88.29
4	Bagged Trees	db1	6	83.07
5	Bagged Trees	db1	6	81.45
6	Bagged Trees	db1	6	78.57

**Table 5 sensors-20-04677-t005:** Comparison of different classifiers in two class classification with different wavelet. The $n_{DSSM}=6$. Highest values are highlighted in boldface.

Accuracy (%)	db1	db2	db3	db4	db5	db6	db8	db16	db32	sym2	sym8	sym16	coif1	coif3	dmey
Linear Discriminant	94.21	93.94	94.33	93.93	93.52	93.17	92.53	91.72	91.66	93.90	92.35	91.68	93.98	92.59	92.24
Quadratic Discriminant	92.94	93.73	94.42	92.89	91.90	91.07	91.15	87.63	87.13	93.68	90.99	88.37	94.14	91.29	87.27
Quadratic SVM	95.13	95.01	95.16	94.88	94.62	94.43	94.14	93.71	93.47	95.09	94.13	93.59	95.12	94.25	93.68
Fine KNN	91.96	90.96	92.62	91.05	90.06	89.82	88.70	87.60	86.58	90.84	88.43	87.00	90.61	89.52	88.22
Bagged Trees	95.71	95.72	95.70	95.59	95.42	95.43	95.16	94.93	95.00	**95.79**	95.04	94.82	95.71	95.29	95.01
RUSBoosted Trees	94.28	94.08	94.68	94.41	94.07	93.94	93.75	93.02	93.08	94.05	93.78	92.79	94.20	93.76	93.20

**Table 6 sensors-20-04677-t006:** The accuracy (%) of the two class sleep stage classification with different wavelet bases and different order of DSSM under R&K standard. Only DSSMFs are used, no LEFs.

		$\omega_{\mathrm{DSSM}}$														
		**db1**	**db2**	**db3**	**db4**	**db5**	**db6**	**db8**	**db16**	**db32**	**sym2**	**sym8**	**sym** **16**	**coif1**	**coif3**	**dmey**
$n_{\mathrm{DSSM}}$	5	95.52	95.55	95.69	95.59	95.17	95.16	95.09	94.98	94.96	95.72	94.94	94.92	95.54	95.07	95.24
	6	95.71	95.72	95.70	95.59	95.42	95.43	95.16	94.93	95.00	**95.79**	95.04	94.82	95.71	95.29	95.01
	7	95.63	95.54	95.57	95.70	95.61	95.38	95.28	94.70	94.80	95.59	95.31	94.58	95.60	95.21	94.94
	8	95.47	95.51	95.51	95.64	95.50	95.23	95.15	94.85	94.47	95.45	95.16	94.75	95.44	95.20	94.62
	9	95.57	95.52	95.54	95.62	95.71	95.34	95.12	94.93	94.33	95.49	95.26	94.55	95.45	95.21	94.45
	10	95.38	95.45	95.46	95.57	95.54	95.28	95.23	94.72	94.12	95.48	95.10	94.76	95.44	95.23	94.32

**Table 7 sensors-20-04677-t007:** The accuracy (%) of three class sleep stage classification with different wavelet bases and different order of DSSM under R&K standard. Only DSSMFs are used, no LEFs.

		$\omega_{\mathrm{DSSM}}$														
		**db1**	**db2**	**db3**	**db4**	**db5**	**db6**	**db8**	**db16**	**db32**	**sym2**	**sym8**	**sym16**	**coif1**	**coif3**	**dmey**
$n_{\mathrm{DSSM}}$	5	87.90	87.92	87.72	88.04	86.70	86.73	86.18	85.50	85.25	87.86	86.11	85.55	87.54	86.03	85.43
	6	**88.29**	88.03	88.26	88.05	87.85	87.70	86.72	85.38	85.25	87.82	86.54	85.20	87.90	87.09	85.65
	7	87.88	87.72	88.13	87.94	87.67	87.76	87.20	84.85	84.27	87.88	87.43	85.00	87.96	87.18	84.82
	8	87.67	87.87	88.10	87.96	87.86	87.68	87.24	85.66	84.15	87.88	87.53	85.83	87.71	87.16	84.20
	9	87.81	87.79	88.07	87.83	87.96	87.81	87.55	85.99	83.85	87.84	87.55	86.37	87.67	87.24	83.79
	10	87.62	87.88	87.84	88.02	87.80	87.78	87.58	86.14	83.93	87.92	87.52	86.62	87.27	87.16	83.53

**Table 8 sensors-20-04677-t008:** The accuracy (%) of four class sleep stage classification with different wavelet bases and different order of DSSM under R&K standard. Only DSSMFs are used, no LEFs.

		$\omega_{\mathrm{DSSM}}$														
		**db1**	**db2**	**db3**	**db4**	**db5**	**db6**	**db8**	**db16**	**db32**	**sym2**	**sym8**	**sym16**	**coif1**	**coif3**	**dmey**
$n_{\mathrm{DSSM}}$	5	82.61	82.69	82.51	82.93	81.36	81.37	80.29	79.83	79.53	82.88	80.59	79.59	82.13	80.49	79.76
	6	**83.07**	82.71	82.76	82.78	82.29	82.23	81.40	79.63	79.16	82.71	81.07	79.43	82.58	81.75	79.79
	7	82.38	82.78	82.81	82.42	82.20	82.35	81.45	78.91	78.37	82.76	81.70	79.03	82.45	81.69	78.62
	8	82.29	82.40	82.69	82.32	82.33	82.06	81.51	79.78	78.02	82.39	81.83	79.69	82.36	81.36	78.28
	9	82.29	82.56	82.54	82.49	82.24	82.25	81.76	80.00	77.68	82.35	81.99	80.35	82.53	81.58	78.00
	10	82.09	82.48	82.71	82.71	82.10	82.38	81.95	79.95	77.70	82.56	81.65	80.50	81.93	81.57	77.48

**Table 9 sensors-20-04677-t009:** The accuracy (%) of five class sleep stage classification with different wavelet bases and different order of DSSM under R&K standard. Only DSSMFs are used, no LEFs.

		$\omega_{\mathrm{DSSM}}$														
		**db1**	**db2**	**db3**	**db4**	**db5**	**db6**	**db8**	**db16**	**db32**	**sym2**	**sym8**	**sym16**	**coif1**	**coif3**	**dmey**
$n_{\mathrm{DSSM}}$	5	81.14	81.15	80.66	81.17	79.79	79.83	79.04	78.55	78.30	81.20	79.27	78.41	80.51	79.30	78.72
	6	**81.45**	81.38	81.42	81.17	81.00	80.60	79.92	78.38	78.13	81.33	79.66	78.15	81.19	80.23	78.71
	7	80.80	80.87	81.00	80.88	80.63	80.48	79.87	77.75	77.08	80.91	80.27	77.53	80.87	80.07	77.61
	8	80.95	81.07	81.08	80.92	80.62	80.52	80.13	78.13	76.89	81.00	80.15	78.48	80.85	80.16	77.20
	9	80.95	80.90	80.75	80.88	80.67	80.56	80.27	78.73	76.72	80.86	79.99	78.93	80.64	79.97	76.78
	10	80.64	80.86	80.97	80.94	80.71	80.72	80.29	78.42	76.62	80.93	80.22	78.94	80.47	79.90	76.12

**Table 10 sensors-20-04677-t010:** The accuracy (%) of six class sleep stage classification with different wavelet bases and different order of DSSM under R&K standard. Only DSSMFs are used, no LEFs.

		$\omega_{\mathrm{DSSM}}$														
		**db1**	**db2**	**db3**	**db4**	**db5**	**db6**	**db8**	**db16**	**db32**	**sym2**	**sym8**	**sym16**	**coif1**	**coif3**	**dmey**
$n_{\mathrm{DSSM}}$	5	78.01	77.96	77.56	77.94	76.47	76.46	75.44	74.99	74.58	77.67	75.80	75.02	77.17	75.58	74.99
	6	**78.57**	78.20	78.26	78.16	77.75	77.56	76.88	75.41	75.07	78.40	76.43	75.02	78.38	77.02	75.34
	7	77.31	77.39	77.52	77.38	76.89	77.00	76.32	74.11	73.63	77.39	76.54	74.22	77.03	76.77	74.09
	8	77.36	77.43	77.47	77.40	76.93	76.84	76.15	74.76	73.07	77.58	76.55	74.75	77.18	76.21	73.54
	9	77.25	77.38	77.56	77.44	76.94	76.89	76.39	74.91	73.01	77.22	76.79	75.21	77.07	76.34	73.10
	10	76.76	77.31	77.66	77.59	76.99	76.89	76.64	74.64	72.88	77.24	76.34	75.22	77.08	76.45	72.47

**Table 11 sensors-20-04677-t011:** The accuracy (%) of two class sleep stage classification with different wavelet bases and different order of DSSM under AASM standard. Only DSSMFs are used, no LEFs.

		$\omega_{\mathrm{DSSM}}$														
		**db1**	**db2**	**db3**	**db4**	**db5**	**db6**	**db8**	**db16**	**db32**	**sym2**	**sym8**	**sym16**	**coif1**	**coif3**	**dmey**
$n_{\mathrm{DSSM}}$	5	95.09	95.01	95.20	94.99	94.63	94.53	94.63	94.67	94.44	95.10	94.40	94.54	94.91	94.54	94.56
	6	95.59	95.63	95.68	95.55	95.42	95.29	95.10	94.75	94.79	**95.73**	95.02	94.79	95.72	95.12	94.90
	7	95.19	95.21	95.50	95.30	95.11	94.89	94.91	94.33	94.26	95.13	94.99	94.35	95.18	95.00	94.30
	8	94.82	95.14	95.09	95.31	94.90	94.73	94.71	94.25	93.83	95.17	94.62	94.16	94.99	94.74	94.03
	9	95.16	95.28	95.28	95.10	95.26	95.04	94.96	94.16	93.79	95.22	94.68	94.25	95.11	94.81	93.97
	10	95.07	95.20	95.18	95.40	95.09	94.98	94.95	94.24	93.41	95.10	94.72	94.20	95.17	95.09	93.46

**Table 12 sensors-20-04677-t012:** The accuracy (%) of three class sleep stage classification with different wavelet bases and different order of DSSM under AASM standard. Only DSSMFs are used, no LEFs.

		$\omega_{\mathrm{DSSM}}$														
		**db1**	**db2**	**db3**	**db4**	**db5**	**db6**	**db8**	**db16**	**db32**	**sym2**	**sym8**	**sym16**	**coif1**	**coif3**	**dmey**
$n_{\mathrm{DSSM}}$	5	87.12	87.04	86.90	87.17	85.81	85.84	85.13	84.83	84.69	86.97	84.81	84.82	86.45	85.45	84.85
	6	**87.52**	87.35	87.55	87.63	87.12	86.88	86.16	84.49	84.61	87.26	85.87	84.48	87.21	86.46	84.85
	7	87.62	87.32	87.83	87.49	87.15	87.30	86.58	84.55	83.95	87.49	87.00	84.38	87.44	86.85	84.18
	8	86.92	87.56	87.59	87.44	87.15	87.26	86.73	85.55	83.23	87.20	86.77	85.10	87.14	86.67	83.53
	9	87.52	87.28	87.68	87.43	87.34	87.43	87.16	85.56	83.36	87.55	86.95	85.74	87.20	86.86	83.07
	10	87.38	87.51	87.52	87.62	87.14	87.46	87.32	85.64	83.19	87.37	87.02	86.09	87.25	86.87	82.79

**Table 13 sensors-20-04677-t013:** The accuracy (%) of four class sleep stage classification with different wavelet bases and different order of DSSM under AASM standard. Only DSSMFs are used, no LEFs.

		$\omega_{\mathrm{DSSM}}$														
		**db1**	**db2**	**db3**	**db4**	**db5**	**db6**	**db8**	**db16**	**db32**	**sym2**	**sym8**	**sym16**	**coif1**	**coif3**	**dmey**
$n_{\mathrm{DSSM}}$	5	81.26	80.82	80.65	80.97	79.27	78.90	78.14	77.80	77.40	80.52	78.48	77.86	80.22	78.13	77.87
	6	**81.51**	80.87	81.18	81.12	80.44	79.86	79.35	77.16	77.01	80.89	79.27	77.35	80.81	79.40	77.56
	7	81.13	80.91	81.40	80.86	80.29	80.21	79.61	76.98	75.92	80.74	80.05	77.19	80.48	79.60	76.52
	8	80.22	80.60	80.34	80.56	80.10	79.67	79.37	77.82	75.32	80.66	79.42	77.89	80.15	79.41	76.05
	9	80.66	80.75	80.77	80.47	80.25	80.38	79.72	77.46	75.54	81.07	79.89	78.11	80.89	79.82	75.01
	10	80.50	80.83	81.06	81.29	80.33	80.43	80.21	77.78	75.52	80.69	79.53	78.34	80.15	79.87	74.99

**Table 14 sensors-20-04677-t014:** The accuracy (%) of five class sleep stage classification with different wavelet bases and different order of DSSM under AASM standard. Only DSSMFs are used, no LEFs.

		$\omega_{\mathrm{DSSM}}$														
		**db1**	**db2**	**db3**	**db4**	**db5**	**db6**	**db8**	**db16**	**db32**	**sym2**	**sym8**	**sym16**	**coif1**	**coif3**	**dmey**
$n_{\mathrm{DSSM}}$	5	78.70	78.49	78.22	78.83	77.07	76.94	76.10	75.16	75.45	78.70	76.34	75.41	77.90	76.14	75.82
	6	**78.72**	78.55	78.54	78.67	77.74	77.73	77.08	75.36	74.99	78.58	77.22	75.15	78.29	77.21	75.72
	7	78.57	78.49	78.38	78.15	77.95	77.77	77.42	74.81	73.54	78.41	77.19	74.64	77.93	77.02	74.52
	8	78.06	78.11	77.94	77.83	77.43	77.56	77.09	75.55	73.45	78.23	77.35	75.33	77.87	77.18	73.64
	9	78.28	78.30	78.30	77.74	77.77	78.03	77.64	75.49	73.27	78.60	77.41	75.88	78.14	77.11	73.29
	10	77.91	78.29	78.26	78.29	77.87	77.83	77.63	75.12	73.25	78.20	77.49	75.55	78.26	77.37	72.38

**Table 15 sensors-20-04677-t015:** The accuracy (%) of two class sleep stage classification with different $\omega_{LE}$ and $l_{LE}$ under R&K standard.

		ωLE														
		**db1**	**db2**	**db3**	**db4**	**db5**	**db6**	**db8**	**db16**	**db32**	**sym2**	**sym8**	**sym16**	**coif1**	**coif3**	**dmey**
$l_{\mathrm{LE}}$	3	95.71	95.69	95.70	95.84	95.68	95.64	95.68	95.71	95.64	95.67	95.66	95.77	95.69	95.70	95.69
	4	95.74	95.69	95.74	95.96	95.69	95.75	95.71	95.63	95.71	95.71	95.58	95.70	95.73	95.56	95.63
	5	95.82	95.89	95.90	**96.17**	95.88	95.87	95.81	95.72	95.85	95.80	95.74	95.74	95.89	95.74	95.72
	6	95.80	95.71	95.93	96.06	95.70	95.73	95.69	95.67	95.58	95.84	95.81	95.63	95.82	95.65	95.49
	7	95.76	95.74	95.64	95.70	95.71	95.57	95.61	95.57	95.60	95.58	95.53	95.72	95.64	95.54	95.51

**Table 16 sensors-20-04677-t016:** The accuracy (%) of three class sleep stage classification with different $\omega_{LE}$ and $l_{LE}$ under R&K standard.

		ωLE														
		**db1**	**db2**	**db3**	**db4**	**db5**	**db6**	**db8**	**db16**	**db32**	**sym2**	**sym8**	**sym16**	**coif1**	**coif3**	**dmey**
$l_{\mathrm{LE}}$	3	88.12	88.07	87.91	88.48	88.12	88.11	88.21	87.86	87.97	88.09	88.02	87.96	88.15	88.10	88.10
	4	88.22	88.22	88.22	88.59	88.81	88.12	87.86	88.08	87.91	88.02	88.05	87.92	88.02	87.85	87.94
	5	88.51	88.56	88.61	88.72	**88.89**	88.75	88.14	88.18	88.09	88.47	88.37	88.14	88.64	88.18	87.98
	6	88.11	88.09	88.00	88.66	88.73	88.00	87.88	87.63	87.54	88.17	87.70	87.84	87.82	87.76	87.43
	7	87.92	87.76	87.58	88.62	88.62	87.65	87.27	87.34	87.35	87.83	87.49	87.39	87.61	87.46	87.27

**Table 17 sensors-20-04677-t017:** The accuracy (%) of four class sleep stage classification with different $\omega_{LE}$ and $l_{LE}$ under R&K standard.

		ωLE														
		**db1**	**db2**	**db3**	**db4**	**db5**	**db6**	**db8**	**db16**	**db32**	**sym2**	**sym8**	**sym16**	**coif1**	**coif3**	**dmey**
$l_{\mathrm{LE}}$	3	82.87	82.76	82.98	83.06	82.99	82.80	82.87	82.98	82.75	82.85	82.82	82.91	82.87	82.97	83.02
	4	82.97	82.91	83.23	83.54	83.02	82.89	82.85	82.86	82.83	82.93	82.92	82.68	82.89	82.79	82.75
	5	83.53	83.52	83.53	**83.97**	83.71	83.25	83.04	82.99	83.18	83.57	83.07	83.07	83.38	82.98	82.84
	6	83.25	82.91	83.67	83.80	82.55	82.76	82.25	82.38	82.40	82.91	82.42	82.11	82.87	82.37	82.17
	7	82.57	82.57	82.52	82.60	82.40	82.29	81.93	81.90	82.03	82.56	82.13	81.92	82.43	81.85	81.89

**Table 18 sensors-20-04677-t018:** The accuracy (%) of five class sleep stage classification with different $\omega_{LE}$ and $l_{LE}$ under R&K standard.

		ωLE														
		**db1**	**db2**	**db3**	**db4**	**db5**	**db6**	**db8**	**db16**	**db32**	**sym2**	**sym8**	**sym16**	**coif1**	**coif3**	**dmey**
$l_{\mathrm{LE}}$	3	81.24	81.42	81.52	81.66	81.45	81.29	81.31	81.28	81.35	81.37	81.19	81.13	81.27	81.32	81.30
	4	81.49	81.55	81.67	81.93	81.70	81.09	81.18	81.10	81.21	81.30	81.45	81.33	81.43	81.28	80.97
	5	81.87	81.66	**82.32**	82.28	81.61	81.42	81.09	80.95	81.01	81.34	80.93	80.92	81.19	81.01	80.79
	6	81.59	81.48	81.29	81.82	81.07	80.98	80.67	80.85	80.75	81.52	80.90	80.89	81.36	80.68	80.44
	7	81.07	81.04	80.89	81.47	80.70	80.38	80.58	80.59	80.56	80.80	80.69	80.44	80.76	80.47	80.55

**Table 19 sensors-20-04677-t019:** The accuracy (%) of six class sleep stage classification with different $\omega_{LE}$ and $l_{LE}$ under R&K standard.

		ωLE														
		**db1**	**db2**	**db3**	**db4**	**db5**	**db6**	**db8**	**db16**	**db32**	**sym2**	**sym8**	**sym16**	**coif1**	**coif3**	**dmey**
$l_{\mathrm{LE}}$	3	78.25	78.39	78.44	78.52	78.18	78.35	77.99	78.08	78.15	78.29	78.06	78.03	78.21	78.14	78.20
	4	78.63	78.64	78.67	78.80	78.67	78.65	78.56	78.54	78.54	78.62	78.44	78.45	78.63	78.48	78.28
	5	78.91	78.91	78.88	**78.92**	78.23	78.42	78.38	78.22	78.38	78.77	78.24	78.20	78.81	78.44	77.98
	6	78.45	78.35	78.26	78.64	77.97	77.90	77.88	77.68	77.78	78.36	77.74	77.57	78.22	77.64	77.36
	7	77.92	77.66	77.48	77.55	77.31	77.22	77.25	77.08	77.63	77.60	77.45	77.18	77.81	77.22	77.02

**Table 20 sensors-20-04677-t020:** The confusion matrix of six classes sleep state classification on DRMS database under the R&K standard. The $l_{LE}=5$, $\omega_{LE}=db4$, $n_{DSSM}=6$, $\omega_{DSSM}=db1$.

		Automatic Classification							
		**Awa**	**REM**	**S1**	**S2**	**S3**	**S4**	**Sen** (%)	**Overall Accuracy (%)**
**Expert**	Awa	5247	96	85	155	3	15	93.68	78.92
	REM	203	3697	86	566	1	2	81.16	
	S1	418	784	257	328	1	0	14.37	
	S2	262	731	62	11852	248	119	89.29	
	S3	20	0	0	1022	543	527	25.71	
	S4	174	0	0	231	271	2395	77.99	

**Table 21 sensors-20-04677-t021:** The accuracy (%) of two class sleep stage classification with different $\omega_{LE}$ and $l_{LE}$ under AASM standard.

		ωLE														
		**db1**	**db2**	**db3**	**db4**	**db5**	**db6**	**db8**	**db16**	**db32**	**sym2**	**sym8**	**sym16**	**coif1**	**coif3**	**dmey**
$l_{\mathrm{LE}}$	3	95.39	95.21	95.34	96.18	95.25	95.25	95.33	95.21	95.26	95.99	95.28	95.21	95.35	95.24	95.30
	4	96.00	95.70	95.84	96.24	95.75	95.37	95.34	95.25	95.34	96.24	95.31	95.24	95.85	95.39	95.43
	5	95.87	95.91	95.94	**96.48**	95.95	95.36	95.41	95.46	95.45	96.41	95.45	95.52	95.53	95.40	95.45
	6	95.61	95.37	95.79	96.16	95.33	95.36	95.37	95.28	95.25	96.34	95.38	95.42	95.32	95.30	95.27
	7	95.42	95.31	95.26	95.90	95.38	95.26	95.24	95.34	95.34	95.35	95.21	95.30	95.38	95.22	95.11

**Table 22 sensors-20-04677-t022:** The accuracy (%) of three class sleep stage classification with different $\omega_{LE}$ and $l_{LE}$ under AASM standard.

		ωLE														
		**db1**	**db2**	**db3**	**db4**	**db5**	**db6**	**db8**	**db16**	**db32**	**sym2**	**sym8**	**sym16**	**coif1**	**coif3**	**dmey**
$l_{\mathrm{LE}}$	3	87.99	87.92	87.62	87.56	87.75	87.67	87.52	87.85	87.55	87.63	87.64	87.77	87.73	87.51	87.59
	4	87.80	87.65	87.67	87.71	87.68	87.64	87.51	87.69	87.77	87.76	87.78	87.70	87.79	87.58	87.88
	5	88.00	**88.26**	88.04	88.22	87.96	87.99	87.85	87.92	87.84	88.17	88.04	88.04	88.04	87.99	87.87
	6	87.89	87.78	87.75	87.71	87.42	87.64	87.59	87.34	87.55	87.64	87.58	87.34	87.82	87.44	87.26
	7	87.13	87.71	87.43	87.92	87.23	87.62	87.38	87.39	87.52	87.75	87.55	87.34	87.51	87.58	87.11

**Table 23 sensors-20-04677-t023:** The accuracy (%) of four class sleep stage classification with different $\omega_{LE}$ and $l_{LE}$ under AASM standard.

		ωLE														
		**db1**	**db2**	**db3**	**db4**	**db5**	**db6**	**db8**	**db16**	**db32**	**sym2**	**sym8**	**sym16**	**coif1**	**coif3**	**dmey**
$l_{\mathrm{LE}}$	3	81.14	81.43	81.45	81.42	81.23	81.40	81.44	81.53	81.33	81.57	81.07	81.55	81.65	81.29	81.15
	4	81.64	81.47	81.36	81.36	81.46	81.33	81.25	81.47	81.37	81.75	81.09	81.31	81.61	81.26	81.37
	5	81.79	82.03	81.91	**82.08**	81.84	81.93	81.59	81.82	81.73	82.23	81.51	81.65	82.12	81.64	81.40
	6	81.83	81.66	81.66	81.76	81.39	81.40	81.37	81.32	80.99	81.51	81.16	81.01	81.85	80.99	80.65
	7	81.35	81.28	81.19	81.47	81.06	80.96	80.93	80.66	80.75	81.37	80.85	80.68	81.40	80.50	80.40

**Table 24 sensors-20-04677-t024:** The accuracy (%) of five class sleep stage classification with different $\omega_{LE}$ and $l_{LE}$ under AASM standard.

		ωLE														
		**db1**	**db2**	**db3**	**db4**	**db5**	**db6**	**db8**	**db16**	**db32**	**sym2**	**sym8**	**sym16**	**coif1**	**coif3**	**dmey**
$l_{\mathrm{LE}}$	3	79.24	79.02	78.87	79.20	79.08	78.99	78.86	78.96	78.77	78.89	79.16	79.09	79.03	78.66	79.09
	4	79.41	79.26	79.33	79.25	79.14	78.80	79.08	79.03	78.91	78.93	78.82	79.16	79.05	79.03	79.01
	5	**79.97**	79.54	79.48	79.90	79.45	79.34	79.36	79.38	79.62	79.77	79.36	79.15	79.53	79.43	79.05
	6	79.10	79.29	79.20	79.23	79.13	78.94	78.89	78.68	78.64	79.20	78.86	78.52	79.39	78.43	78.01
	7	79.04	78.69	78.89	78.85	78.67	78.39	78.48	78.33	78.16	78.98	78.71	78.22	78.66	78.30	77.67

**Table 25 sensors-20-04677-t025:** The confusion matrix of five classes sleep state classification on DRMS database under the AASM standard. The $l_{L}E=5$, $\omega_{LE}=db4$, $n_{DSSM}=6$, $\omega_{DSSM}=db1$.

		Automatic Classification						
		**Awa**	**REM**	**N1**	**N2**	**N3**	**Sen(%)**	**Overall Accuracy (%)**
**Expert**	Awa	3306	53	68	111	21	92.89	79.90
	REM	131	2452	93	330	13	81.22	
	N1	341	480	260	389	10	17.57	
	N2	229	499	50	7056	417	85.52	
	N3	77	1	0	761	3117	78.79	

**Table 26 sensors-20-04677-t026:** The classification accuracy and Cohen’s Kappa Coefficient of 2–6 class sleep classification on S-EDF database.

	6 Classes	5 Classes	4 Classes	3 Classes	2 Classes
Accuracy	92.04%	92.50%	93.87%	94.90%	98.74%
Cohen’s Kappa Coefficient	0.8266	0.8364	0.8646	0.8834	0.9697

**Table 27 sensors-20-04677-t027:** the Confusion matrix of six classes sleep state classification on S-EDF database.

		Automatic Classification							
		**Awa**	**REM**	**S1**	**S2**	**S3**	**S4**	**Sen** (%)	**Overall Accuracy (%)**
**Expert**	Awa	73165	483	46	141	0	0	99.09	92.04
	REM	876	4819	61	988	0	0	71.46	
	S1	578	1174	583	682	0	0	19.32	
	S2	375	863	76	15631	254	50	90.62	
	S3	71	0	0	1003	1055	159	46.11	
	S4	23	0	0	171	256	1060	70.20	

**Table 28 sensors-20-04677-t028:** The classification accuracy and Cohen’s Kappa Coefficient of 2–5 class sleep classification on ISRUC3 database.

Classes	5 Classes	4 Classes	3 Classes	2 Classes
accuracy	81.65%	84.68%	90.54%	96.18%
Cohen’s Kappa Coefficient	0.7629	0.7729	0.8112	0.878

**Table 29 sensors-20-04677-t029:** The confusion matrix for five classes case on ISRUC3 database.

		Automatic Classification						
		**Awa**	**REM**	**N1**	**N2**	**N3**	**Sen(%)**	**Overall Accuracy (%)**
**Expert**	Awa	1537	13	84	53	15	90.31	81.65
	REM	36	1032	89	68	13	83.36	
	N1	91	135	648	242	7	57.70	
	N2	68	85	128	2312	257	81.12	
	N3	15	1	2	229	1729	87.50	

**Table 30 sensors-20-04677-t030:** The accuracy comparison of various published methods on DRMS database under the R&K standard. Highest accuracy in each case is highlighted in bold.

	Epoch Mumber	6 Classes (%)	5 Classes (%)	4 Classes (%)	3 Classes (%)	2 Classes (%)	Cross-Validation
Hassan et al. [3]	30401	70.73	73.50	79.12	84.4	93.3	10-fold
Hassan et al. [11]	30401	68.74	73.05	78.8	82.96	94.02	0.5/0.5
Shen et al. [27]	30401	78.2	80.9	82.7	87.7	94.9	10-fold
Proposed method Without LEFs	30401	78.52	81.26	82.81	87.95	95.59	10-fold
Proposed method with IMBEFs	30401	**78.92**	**82.28**	**83.97**	**88.72**	**96.17**	10-fold

**Table 31 sensors-20-04677-t031:** The accuracy comparison of various published methods on the Dreams Subjects database under the AASM standard. Highest accuracy in each case is highlighted in bold.

	Epoch number	5 Classes (%)	4 Classes (%)	3 Classes (%)	2 Classes (%)	Cross-validation
Hassan et al. [3]	20265	72.28	79.44	83.75	95.2	10-fold
Hassan et al. [11]	20265	74.59	80.0	85.42	**97.2**	10-fold
Ghimatgar et al. [7]	20265	78.08	80.38	86.88	94.8	20-fold
Proposed Method Without LEFs	20265	78.72	80.9	87.52	95.7	10-fold
Proposed Method with IMBEFs	20265	**79.90**	**82.08**	**88.22**	96.48	10-fold

**Table 32 sensors-20-04677-t032:** The accuracy comparison of various published methods on the S-EDF database under the R&K standard. Highest accuracy in each case is highlighted in bold.

	Epoch Number	6 Classes (%)	5 Classes (%)	4 Classes (%)	3 Classes (%)	2 Classes (%)	Cross-Validation
Hassan et al. [3]	15188	90.38	91.50	92.11	94.8	97.5	0.5/0.5
Abdulla et al. [6]	23806	**93**	–	–	–	–	-
Ghimatgar et al. [7]	15188	89.91	91.11	92.19	94.65	98.19	0.5/0.5
Ghimatgar et al. [7]	40100	79.13	81.86	83.71	88.39	95.98	0.5/0.5
Hassan et al. [10]	15188	88.62	90.11	91.2	93.55	97.73	0.5/0.5
Hassan et al. [11]	15188	88.07	83.49	92.66	94.23	98.15	0.5/0.5
Sharma et al. [12]	15139	90.03	91.13	92.29	94.66	98.02	10-fold CV
Michielli et al. [17]	10280	–	86.7	–	–	–	10-fold CV
Shen et al. [27]	103505	91.9	92.3	93.0	93.9	98.6	10-fold CV
Sharma et al. [28]	85900	91.5	91.7	92.1	93.9	98.3	10-fold CV
Liang et al. [29]	3708	–	83.6	–	–	–	0.5/0.5
Hsu et al. [30]	2880	–	87.2	–	–	–	10-fold CV
Hassan et al. [31]	15188	89.6	90.8	91.6	93.9	97.2	0.5/0.5
Zhu et al. [32]	14963	87.5	88.9	89.3	92.6	97.9	10-fold CV
Jiang et al. [33]	36972	–	91.5	–	–	–	2-fold CV
Rahman et al. [34]	15188	90.26	91.02	92.89	94.1	98.24	0.5/0.5
Supratak et al. [35]	41950	–	79.8	–	–	–	20-fold CV
Proposed Method	104368	92.04	**92.50**	**93.87**	**94.90**	**98.74**	10-fold CV

**Table 33 sensors-20-04677-t033:** The accuracy comparison of the ISRUC3 database with the AASM standard. Highest accuracy in each case is highlighted in bold.

		Epoch number	5 Classes	4 Classes	3 Classes	2 Classes
Overall Accuracy	Ghimatgar et al. [7]	8889	77.56	82.74	88.26	93.76
	Proposed Method	8889	**81.65**	**84.68**	**90.54**	**96.18**
Cohen’s kappa Coefficient	Ghimatgar et al. [7]	8889	0.71	0.75	0.77	0.79
	Proposed Method	8889	**0.7629**	**0.7729**	**0.8112**	**0.878**

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
