# Peer review of "An Automatic Sleep Stage Classification Algorithm Using Improved Model Based Essence Features"

_sensors, 2020, doi:10.3390/s20174677_

Round 1

Reviewer 1 Report

The article describes the topic of EEG sleep staging using different classifiers. The article is written in a good English, is clear and interesting. Although its originality
could be arguable, I think it should be published after considering some of my comments bellow.

1.
The 10-fold crossvalidation with random distribution of instances into the folds could be slightly optimistically biased. It happens that two instances that are close to each other (in time) can be in both training and testing set). This however cannot happen in reality where the classifier is used at least couple of days after its training data were collected. On the other hand, a block based hold-out method could be pessimistically biased. I would recommend to use one-subject-out CV as described in the next comment.

2.
It is not clear if the classifier was used interpersonally, i.e. it is supposed that it will be used on subjects that differ from the subjects that were used for training data acquisition. This should be clearly stated in sec. 3. In such a case, the correct approach is to use one-subject-out type of crossvalidation, which is more realistic and less biased. Nevertheless, I believe that the optimistic bias will not be so critical and do not ask for re-computation of the results. However, this should be clearly described and commented in the text (e.g. at the beginning of sec. 3).

3.
It is not clear how accuracy is computed. A definition and a proper equation should be added. If a percentage of correctly classified instances is used, it should be accompanied by class-based accuracies/sensitivities to show how the minor classes are classified and prevent the situation, where one minority class is absolutely missclassified. This happens e.g. in table 20 with classes S1 and S3. The discussion section and comparison to state-of-the-art should contain also class sensitivities. The high accuracy simply does not mean that the classifier is good for sleep staging. I would recommend to use the class sensitivity averaged over all classes. Especially, it should be used in the tuning phase. However, I appreciate the discussion on page 17 which is touching this topic.

Author Response

Response to Reviewer 1 Comments

Point 1: The 10-fold cross-validation with random distribution of instances into the folds could be slightly optimistically biased. It happens that two instances that are close to each other (in time) can be in both training and testing set). This however cannot happen in reality where the classifier is used at least couple of days after its training data were collected. On the other hand, a block based hold-out method could be pessimistically biased. I would recommend to use one-subject-out CV as described in the next comment.

Response 1: We really appreciate your expertise. Leave one-subject-out cross validation(LOOCV)is indeed a good verification method. We will use the LOOCV verification method on the basis of the k-fold CV method in the follow-up research.

Point 2: It is not clear if the classifier was used interpersonally, i.e. it is supposed that it will be used on subjects that differ from the subjects that were used for training data acquisition. This should be clearly stated in sec. 3. In such a case, the correct approach is to use one-subject-out type of cross-validation, which is more realistic and less biased. Nevertheless, I believe that the optimistic bias will not be so critical and do not ask for re-computation of the results. However, this should be clearly described and commented in the text (e.g. at the beginning of sec. 3).

Response 2: Many thanks for your helpful and valuable comments. As you have commented, in reality, the trained classifiers are mostly trained by other people's data. When classifying the stages on new subjects, the classification results may not be accurate enough. In this paper, we use multiple databases. Each database has more or less subjects, and the samples of individual subjects are unevenly distributed. So for the convenience of verification, k-fold CV is uniformly adopted. However, in future work, in addition to using k-fold CV, we will also use LOOCV to better evaluate the performance of our algorithm. We have described the selection of verification methods in detail in section 3  (from line 187 to line 191). Thank you very much for your comments, which greatly improved the quality of our papers. 

Point 3: It is not clear how accuracy is computed. A definition and a proper equation should be added. If a percentage of correctly classified instances is used, it should be accompanied by class-based accuracies/sensitivities to show how the minor classes are classified and prevent the situation, where one minority class is absolutely misclassified. This happens e.g. in table 20 with classes S1 and S3. The discussion section and comparison to state-of-the-art should contain also class sensitivities. The high accuracy simply does not mean that the classifier is good for sleep staging. I would recommend to use the class sensitivity averaged over all classes. Especially, it should be used in the tuning phase. However, I appreciate the discussion on page 17 which is touching this topic.

Response 3: Many thanks for your positive and valuable comments. It is our mistake that we did not clearly describe the definition of how the classification accuracy is calculated. We have added the equation (14) and equation (15) in Section 3 (from line 198 to line 200) to describe the definition of accuracy and Cohen’s Kappa Coefficient. We really appreciate for your careful reviewing of our manuscript.   

Reviewer 2 Report

  • Several minor grammar errors should be eliminated (i.e, "Where" at the Line 132, ... should not be capitalize the first letter).
  • Regarding Table 32, the work of Shen et al. [32] only reported results with one decimal, how can you conclude that your obtained results are better than their (at column of 2 Classes). You should bold both.
  • In the study, the authors used 198 classifiers including Linear Discriminant, Quadratic Discriminant, Quadratic SVM, Fine KNN, 199 Bagged Trees and RUSBoosted Trees. How about other kinds of classifiers. The authors should mention why they choose these classifiers.

Author Response

Response to Reviewer 2 Comments

Point 1: Several minor grammar errors should be eliminated (i.e., "Where" at the Line 132, ... should not be capitalize the first letter).

Response 1: We appreciate for your careful reviewing of our manuscript. The grammar errors have been eliminated at the line 132, 143,147 and 176.

Point 2: Regarding Table 32, the work of Shen et al. [32] only reported results with one decimal, how can you conclude that your obtained results are better than their (at column of 2 Classes). You should bold both.

Response 2: Thank you very much for your careful review. This is caused by an error when we filled out this form. From the Table 26, we can see that the correct data is 98.74%. And the data in this table 32 (at column of 2 Classes) has been corrected. Very sorry for any inconvenience we have caused you.

Point 3: In the study, the authors used classifiers including Linear Discriminant, Quadratic Discriminant, Quadratic SVM, Fine KNN, Bagged Trees and RUSBoosted Trees. How about other kinds of classifiers. The authors should mention why they choose these classifiers.

Response 3: Many thanks for your helpful and valuable comments. Currently commonly used classifiers are Decision Trees, Discriminant analysis, naive Bayes classifier, support vector machine, nearest neighbor classifier and Ensemble classifier. Combining the distribution characteristics of the number of samples in each sleep stage and the extracted feature properties, the above classifiers are initially screened. Because Decision Trees has poor classification effect when the sample distribution is unbalanced, it is excluded. Naive Bayes classifier is mainly used for text classification, so it is not suitable for the classification of sleep stage in this paper. Therefore, among the remaining kinds of classifiers, typical classifiers will be selected including the Linear Discriminant, Quadratic Discriminant, Quadratic SVM, KNN, Bagged Trees and RUSBoosted Trees. And the reasons for choosing these classifiers have been supplemented in setction3.1 (from line 203 to line 204).

Reviewer 3 Report

1. Please outline some of the practical applications and limitations of the work

2. Do the authors have any plans to make the code and data-set open-source?

3. What is the statistical power of the study?

4. What are the significant advantage of the proposed method against previously published ones?

5. Please provide performance comparison with the following papers in a performance comparison table. 

- "Sleep stage classification using single-channel EOG" in Computers in biology and medicine

- "Computer-aided sleep staging using complete ensemble empirical mode decomposition with adaptive noise and bootstrap aggregating" in Biomedical Signal Processing and Control

- "Automatic sleep scoring using statistical features in the EMD domain and ensemble methods" in Biocybernetics and Biomedical Engineering

- "An automated method for sleep staging from EEG signals using normal inverse Gaussian parameters and adaptive boosting" in Neurocomputing

- "A decision support system for automatic sleep staging from EEG signals using tunable Q-factor wavelet transform and spectral features" in Journal of neuroscience methods

- "A decision support system for automated identification of sleep stages from single-channel EEG signals" in Knowledge-Based Systems

- "Automated identification of sleep states from EEG signals by means of ensemble empirical mode decomposition and random under sampling boosting" in Computer Methods and Programs in Biomedicine

- "Sleep stage classification using single-channel EOG" in Computers in Biology and Medicine

- "Automatic sleep stage classification" in 2015 2nd International Conference on Electrical Information and Communication Technology (EICT)

- "On the classification of sleep states by means of statistical and spectral features from single channel electroencephalogram" in 2015 International Conference on Advances in Computing, Communications and Informatics (ICACCI)

- "Automatic classification of sleep stages from single-channel electroencephalogram" in 2015 Annual IEEE India Conference (INDICON)

- "Dual tree complex wavelet transform for sleep state identification from single channel electroencephalogram" in 2015 IEEE International Conference on Telecommunications and Photonics (ICTP)

6. Does the method work for both R&K and AASM scoring rules? 

7. How did the authors select the training and testing data? They should be randomly selected and mean and SD of performance metrics must be reported. 

8. The authors should cite some recent works on EEG and other biomedical signal processing that to further motivate the need of the work. In this regard, the author must cite the following important papers. 

- "Computer-aided obstructive sleep apnea detection using normal inverse Gaussian parameters and adaptive boosting" in Biomedical Signal Processing and Control

- "Computer-aided obstructive sleep apnea screening from single-lead electrocardiogram using statistical and spectral features and bootstrap aggregating" in Biocybernetics and Biomedical Engineering

- "Computer-aided obstructive sleep apnea identification using statistical features in the EMD domain and extreme learning machine" in Biomedical Physics & Engineering Express

- "Automatic screening of obstructive sleep apnea from single-lead electrocardiogram" in 2015 international conference on Electrical engineering and information communication technology (ICEEICT)

- "A comparative study of various classifiers for automated sleep apnea screening based on single-lead electrocardiogram" in 2015 International Conference on Electrical & Electronic Engineering (ICEEE)

- "Identification of Sleep Apnea from Single-Lead Electrocardiogram" in CSE (http://ieeexplore.ieee.org/abstract/document/7982270/)

- "An expert system for automated identification of obstructive sleep apnea from single-lead ECG using random under sampling boosting" in Neurocomputing

- "Computer-aided sleep apnea diagnosis from single-lead electrocardiogram using dual tree complex wavelet transform and spectral features" in 2015 International Conference on Electrical & Electronic Engineering (ICEEE)

- "Epileptic seizure detection in EEG signals using tunable-Q factor wavelet transform and bootstrap aggregating" in Computer Methods and Programs in Biomedicine.

- "Automatic identification of epileptic seizures from EEG signals using linear programming boosting" in Computer Methods and Programs in Biomedicine.

- "Epilepsy seizure detection using complete ensemble empirical mode decomposition with adaptive noise"

- "Epilepsy seizure detection using complete ensemble empirical mode decomposition with adaptive noise" in Knowledge-Based Systems

- 'Epilepsy and seizure detection using statistical features in the complete ensemble empirical mode decomposition domain' in IEEE TENCON

- "Computer-aided gastrointestinal hemorrhage detection in wireless capsule endoscopy videos" in Computer methods and programs in biomedicine

- "Identification of motor imagery movements from eeg signals using dual tree complex wavelet transform" in 2015 International Conference on Advances in Computing, Communications and Informatics (ICACCI)

- "Motor imagery movements classification using multivariate EMD and short time Fourier transform" in 2015 Annual IEEE India Conference (INDICON)

- "An overview of brain machine interface research in developing countries: Opportunities and challenges" in ICIEV

- "Effect of photic stimulation for migraine detection using random forest and discrete wavelet transform" in Biomedical Signal Processing and Control

- "Sigmoid Wake Probability Model for High-Resolution Detection of Drowsiness Using Electroencephalogram" in EMBC

- "Developing a System for High-Resolution Detection of Driver Drowsiness Using Physiological Signals"

Author Response

Response to Reviewer 3 Comments

Point 1: Please outline some of the practical applications and limitations of the work

Response 1: Thank you for your careful review. Our proposed method is still in the algorithm research stage and will be used in wearable sleep monitoring devices in the future. In the future research, the classification accuracy of S1 stage will be further improved.

Point 2: Do the authors have any plans to make the code and data-set open-source?

Response 2: We really appreciate your expertise. Since this is a cooperative project involving patents and product core algorithms, there is no plan to open code and related dataset temporarily.

Point 3: What is the statistical power of the study?

Response 3: Many thanks for your helpful and valuable comments. The statistical power calculated under different databases is different. However, in the several databases used in this article, the statistical power of the Awake is higher than 0.9. The statistical power of the REM is between 0.71 and 0.82. The statistical power of N1 or S1 is between 0.14 and 0.58. The statistical power of N2 or S2 is between 0.8 and 0.9. The statistical power of the N3 stage is between 0.78 and 0.88. The statistical power of S3 stage is between 0.25 and 0.47. The statistical power of S4 stage is between 0.70 and 0.78.

Point 4: What are the significant advantage of the proposed method against previously published ones?

Response 4: We appreciate for your careful reviewing of our manuscript. In this paper, the improved model based essence features (IMBEFs) combined the locality energy (LE) and dual state space models (DSSM) which are calculated based on the WPD of the EEG epochs.  Compared with those previously published, the method proposed in this paper has achieved higher classification accuracy on the S-EDF database, DMRS database and ISRUC3 database. In addition, it can be seen from the experimental results on multiple databases that the method proposed in this paper has better robustness and generalization.

Point 5: Please provide performance comparison with the following papers in a performance comparison table. 

 Response 5: Thank you very much for your review. The literatures you have given are very informative, and we have been already analyzed and compared the representative results of these papers in the Table 30, Table 31 and Table 32.

Point 6: Does the method work for both R&K and AASM scoring rules? 

 Response 6: Thank you very much for your review. In addition to the S-EDF database and DRMS database under the R&K standard, this paper also adopted the DRMS database and ISRUC3 database under the AASM standard. It can be seen from the experimental results that the method proposed in this paper can work for both AASM and R&K scoring rules.

Point 7: How did the authors select the training and testing data? They should be randomly selected and mean and SD of performance metrics must be reported. 

Response 7: Thank you very much for your valuable comments. The 10-fold cross-validation method is used for all experiments in this paper. In 10-fold cross-validation, the original sample is randomly partitioned into 10 equal size subsamples. Of the 10 subsamples, a single subsample is retained as the validation data for testing the model, and the remaining 9 subsamples are used as training data. The cross-validation process is then repeated 10 times, with each of the 10 subsamples used exactly once as the validation data. The 10 results from the folds can then be averaged to produce a single estimation. The advantage of this method is that all observations are used for both training and validation, and each observation is used for validation exactly once. And The performance metrics have been added as the equation (14) and equation (15) (from line 198 to line 200).

Point 8: The authors should cite some recent works on EEG and other biomedical signal processing that to further motivate the need of the work. In this regard, the author must cite the following important papers. Response 8: Thank you very much for your valuable comments. The references you gave are indeed very useful for our research. Because of space limitations, after careful analysis and research on these papers, we have already cited the representative of these papers. Thank you very much for your recommendation.

Reviewer 4 Report

In this paper, authors have proposed a method for sleep stages classification. The objective of this paper is clear and it is written well. I have a few comments.

1. Please mention the advantages and disadvantages of the work.

2. Do the authors see this technique being useful at some stage at the patient-clinician level or basic science sleep research level?

3. if possible,   ask for professional proofreading.

Author Response

Response to Reviewer 4 Comments

Point 1: Please mention the advantages and disadvantages of the work.

Response 1: We appreciate for your careful reviewing of our manuscript. In this paper, the improved model based essence features (IMBEFs) combined the locality energy (LE) and dual state space models (DSSM) which are calculated based on the WPD of the EEG epochs.  Compared with those previously published, the method proposed in this paper has achieved higher classification accuracy on the S-EDF database, DMRS database and ISRUC3 database. In addition, it can be seen from the experimental results on multiple databases that the method proposed in this paper has better robustness and generalization. And the classification accuracy of S1 and S3 stage will be further improved in future research. 

Point 2: Do the authors see this technique being useful at some stage at the patient-clinician level or basic science sleep research level?

Response 2: Thank you very much for your valuable comments. We currently tend to use this technology for wearable devices. After the subsequent algorithm is improved, we will seek some clinical cooperation.

Point 3: if possible, ask for professional proofreading.

Response 3: Many thanks for your valuable comments. We will cooperate with professional clinical laboratories after we complete the improvement of algorithm performance in future work.
